# Conformational heterogeneity in human interphase chromosome organization reconciles the FISH and Hi-C paradox

Guang Shi [ID] [1] & D. Thirumalai[2]

Hi-C experiments are used to infer the contact probabilities between loci separated by varying genome lengths. Contact probability should decrease as the spatial distance between two loci increases. However, studies comparing Hi-C and FISH data show that in some cases the distance between one pair of loci, with larger Hi-C readout, is paradoxically larger compared to another pair with a smaller value of the contact probability. Here, we show that the FISH-Hi-C paradox can be resolved using a theory based on a Generalized Rouse Model for Chromosomes (GRMC). The FISH-Hi-C paradox arises because the cell population is highly heterogeneous, which means that a given contact is present in only a fraction of cells. Insights from the GRMC is used to construct a theory, without any adjustable parameters, to extract the distribution of subpopulations from the FISH data, which quantitatively reproduces the Hi-C data. Our results show that heterogeneity is pervasive in genome organization at all length scales, reflecting large cell-to-cell variations.

[1] Biophysics Program, Institute for Physical Science and Technology, University of Maryland, College Park, MD 20742, USA. [2] Department of Chemistry, University of Texas at Austin, Austin, TX 78712, USA. Correspondence and requests for materials should be addressed to D.T. (email: dave.thirumalai@gmail.com)

Through remarkable Hi-C experiments[1–6], based on the Chromosome Conformation Capture (3C) technique[7], indirect glimpses of how the genome in a number of species is organized is starting to emerge. Because chromosome lengths are extremely large, ranging from tens of million base pairs in yeast to billion base pairs in human cells, they have to fold into highly compact structures in order to be accommodated in the cell nucleus. This requires that loci that are well separated along the one-dimensional genome sequence be close in three-dimensional (3D) space, which is made possible by forming a large number of loops. The high-throughput Hi-C technique and its variants are used to infer the probability of genome-wide contact formation between loci. In order to determine the contact probabilities between various loci in a genome, Hi-C experiments are performed in an ensemble of millions of cells. The readouts of the Hi-C experiment are contact frequencies between a large number of loci from instantaneous snapshots of each cell, which are then used to construct the contact maps (Hi-C maps). The contact map is a matrix (2D representation) in which the elements represent the probability of contact between two loci that are separated by a specified genomic distance. A high contact count between two loci means that they interact with each other more frequently compared with ones with low contact count.

A complementary and potentially a more direct way to determine genome organization is to measure spatial distances between loci using a low-throughput fluorescence in situ hybridization (FISH) technique[8,9]. In addition to providing 3D distances in fixed cells, recently developed CRISPR–dCas9 FISH can be used to assay the dynamic behavior of loci in real time[10–12]. However, due to the current limitation of the number of distinct color probes, this method provides distance distribution information for only a small number of loci.

FISH and Hi-C, which are entirely different experimental techniques, provide data on different aspects of genome organization. As noted in recent reviews[13,14], there are problems associated with each method. It is difficult to reconcile Hi-C and FISH data for the following reasons. In interpreting the Hi-C contact map, one makes the intuitive assumption that loci with high probability contact must also be spatially close. However, it has been demonstrated using Hi-C and FISH data on the same chromosome that high contact frequency does not always imply proximity in space[13,15–17]. It should be noted that in most cases, the Hi-C and FISH measurements agree very well[8,9,18,19]. However, from a purely theoretical perspective, even a single contradiction is intriguing if the experimental errors can be ruled out. An outcome of our theory is that the discordance between FISH and Hi-C data arises because of extensive heterogeneity, which is embodied by the presence of a variety of conformations adopted by chromosomes in each cell. There are a variety of reasons, including differing fixation conditions and presence of two or more subpopulation of cells in which the chromosomes are present in distinct conformations, which could give rise to the discordance between FISH and Hi-C data, as lucidly described recently[13,14]. Contact between two loci could be a rare event, not present in all cells, which is captured in a Hi-C experiment by performing an ensemble average. We show using a precisely solvable model that due to the absence of a contact between two specific loci in a number of cells, those with higher contact frequency could be spatially farther on an average than two others with lower contact frequency. In contrast, the probability of contact formation using the FISH method can only be obtained if the tail (small distance) of the distance distribution between locus $i$ and $j$ can be accurately measured. For a variety of reasons, including the size of the probe and the signal strength, this not altogether straightforward using the FISH technique. Thus, in order to combine the data from the two powerful techniques, it is crucial to establish a theoretical basis with potential a practical link, between the contact probability and average spatial distance.

Setting aside the conditions under which FISH and Hi-C are performed (see recommendations for comparing the results from the two techniques with minimum bias which are described elsewhere[13]) insights into the discordance between the two methods, when they occur, can be obtained using polymer physics concepts. Recently, Fudenberg and Imakaev[15] performed polymer simulations using a strong attractive energy between two labeled loci and a tenfold weaker interaction between two other loci that are separated by a similar genomic distance. In addition, they also reported simulations based on the loop extrusion model. Both these types of simulations showed there could be discordance between FISH and Hi-C, which we refer to as the FISH–Hi-C paradox. However, they did not provide any solution to the paradox, which is the principal goal of this work.

In addition, recent single-cell Hi-C[20–22] and FISH experiments[8,9,18,19] have revealed that there are substantial cell-to-cell variations on genome organization. However, how to utilize the data reported in these experiments to enhance our understanding of 3D genome structural heterogeneity has not been unexplored. One approach is to create an appropriate polymer model based on Hi-C and imaging data, which would readily allow us to probe the structural variability using simulations[23–26]. Indeed, it has been shown, using Hi-C and FISH data as well as simulations[26], that if the conformation of the chromatin fiber is taken to be homogeneous then trends observed in the FISH data could not be predicted. However, using simulations and including two levels of chromatin organization (open and compact) qualitative trends observed in the FISH data could be recovered[26].

Here, we first establish a relationship between the contact probability and the mean spatial distance using an analytically solvable Generalized Rouse Chromosome Model (GRMC), which incorporates the presence of CTCF/cohein-mediated loops. The GRMC may be thought of as an ideal chromosome model, very much in the spirit of the Rouse model for polymers, in which conceptual issues such as the origin of the FISH–Hi-C paradox can be rigorously established. We first consider the solvable homogeneous limit, in which contacts are present in all the cells. In this case, precise numerical and analytical results show that there is a simple relation between the contact probability, $P$, and the ensemble mean 3D distance $\langle R \rangle$. However, the unavoidable heterogeneity in the cell populations in Hi-C experiments results in contacts between loci only in a fraction of cells. We first show that a direct consequence of the heterogeneity in both GRMC and chromosomes is that two loci ($m$ and $n$) that have higher probability ($P_{mn}$) of being in contact relative to another two loci ($k$ and $l$) does not imply a direct spatial correlation, a finding that has already been qualitatively established in previous studies[13,15]. In other words, the average spatial distance between $m$ and $n$ ($\langle R_{mn} \rangle$) could be larger than $\langle R_{kl} \rangle$, the distance between loci $k$ and $l$, even if $P_{mn} > P_{kl}$. These results provide a basis for understanding the origin of the FISH–Hi-C paradox.

We develop a fully theoretical approach, which allows us to provide quantitative insights into the extent of heterogeneity in chromosome organization. From our theory, it follows that the resolution of the FISH–Hi-C paradox requires invoking the notion of heterogeneity, which implies multiple populations of chromosomes coexist. By using the concepts that emerge from the study of the GRMC, we demonstrate that the information of cell subpopulations can be extracted by fitting the experimental FISH data using our theory, thus allowing us to calculate the Hi-C contact probabilities from the theoretically calculated cumulative distribution function of spatial distance (CDF)—a quantity that can be measured using FISH and super-resolution imaging

methods. Our approach provides a theoretically based method to combine the available FISH and Hi-C data to produce a more refined characterization of the heterogeneous chromosome organization than is possible by using data from just one of the techniques. In other words, sparse data from both the experimental methods can be simultaneously harnessed to predict the 3D organization of chromosomes.

## Results

**Relating contact probability to mean spatial distance.** The exact relationship between $P_{mn}$ (contact probability between $m$th and $n$th locus) and the corresponding mean spatial distance, $\langle R_{mn} \rangle$ for GRMC (see the Methods section for details of the derivation) is,

$$P_{mn} = \mathrm{erf}\left(\frac{2r_c}{\sqrt{\pi}\langle R_{mn} \rangle}\right) - \frac{4}{\pi}\frac{r_c}{\langle R_{mn} \rangle}e^{-\frac{4r_c^2}{\pi\langle R_{mn} \rangle^2}} \equiv R_0(\langle R_{mn} \rangle). \quad (1)$$

The inverse of $R_0(\langle R_{mn} \rangle)$, the solution to Eq. (1), gives the mean spatial distance $\langle R_{mn} \rangle$ as a function of the contact probability $P_{mn}$. Note that $m$ and $n$ are arbitrary locations of any two loci, and thus Eq. (1) is general for any pair of loci.

A couple of conclusions, relevant to the application to the chromosomes, follow from Eq. (1). (i) Note that Eq. (1) is an exact one-to-one relation between the mean distance $\langle R_{mn} \rangle$ and the contact probability $P_{mn}$ provided $r_c$ is known, and if the contacts are present in all the cells, which is not the case in experiments. For small $P_{mn}$, it is easy to show from Eq. (1) that $\langle R_{mn} \rangle \approx r_c P_{mn}^{-1/3}$. For the ideal GRMC, this implies that for any $m$, $n$, $k$, $l$, if $P_{mn} < P_{kl}$ then $\langle R_{mn} \rangle > \langle R_{kl} \rangle$, a consequence anticipated on intuitive grounds. (ii) If the value of the contact probability $P$ and the threshold distance $r_c$ are known precisely, then the distribution of the spatial distance can be readily computed by solving Eq. (1) numerically. In Fig. 1b, we show the comparison between theory (Eq. (1)) and simulations (see the Methods section for details). The simulated curves are computed as follows: first collect $(P_{mn}, \langle R_{mn} \rangle)$ for every pair labeled $(m, n)$, where $P_{mn}$ and $\langle R_{mn} \rangle$ are computed using Eqs. (17) and (18) in the Methods section. The total number of pairs is $N(N-1)/2$. We then bin the points over the values of $P_{mn}$. Finally, the mean value of $\langle R_{mn} \rangle$ for each bin, $\langle R \rangle = E[\langle R_{mn} \rangle]$, is computed where $E[\cdots]$ is the binned average, which is computed using $(1/N_i)\sum_{j=1}^{N_i}\langle R_{mn} \rangle^j$, where $N_i$ is the number of points in the $i$th bin. The bin size, $\Delta$, is centered at $P_{mn}$, spanning $P_{mn} - \Delta/2 \leq P_{mn} \leq P_{mn} + \Delta/2$. Using

this procedure, we find (Fig. 1) that the theory and simulations are in perfect agreement, which validates the theoretical result.

**Contact distance $r_c$ affects the inferred value of the spatial distance.** However, in practice, the elements $P_{mn}$ are measured with (unknown) statistical errors, and the value of the contact threshold $r_c$ is only estimated. In the Hi-C experiments, contact probabilities and $r_c$ by implication, are determined by a series of steps that start with cross-linking spatially adjacent loci using formaldehyde, chopping the chromatin into fragments using restriction enzymes, ligating the fragments with biotin, followed by sequence matching using deep-sequencing methods[5]. Because of the inherent stochasticity associated with the overall Hi-C scheme, as well as the unavoidable heterogeneity (only a fraction of cells has a specific contact and the contact could be dynamic) in the cell population, the relationship $P_{mn}$ and $\langle R_{mn} \rangle$ is not straightforward.

To illustrate how the uncertainty in $r_c$ affects the determination of the spatial distance in GRMC even when the population is homogeneous (all cells have a specific contact), we plot the distributions of distance for $r_c = 0.02, 0.03\,\mu m$ in Fig. 1c. A small change in $r_c$ (from 0.02 to 0.03 μm) completely alters the distance distribution $P(R)$, and hence the mean spatial distance (from ≈0.2 to ≈0.3 μm). For the exactly solvable GRMC, this can be explained by noting that $\langle R_{mn} \rangle \approx r_c P_{mn}^{-1/3}$ for small $P_{mn}$. Because $P_{mn}$ appears in the denominator, any uncertainty in $r_c$ is amplified by $P_{mn}$, especially when $P_{mn}$ is small.

**Heterogeneity causes paradox between FISH and Hi-C.** The expectation that the contact probability should decrease as the mean distance between the loci increases, which is the case in the exactly solvable ideal GRMC ($P_{mn} \approx r_c \langle R_{mn} \rangle^{-3}$), is sometimes violated when the experimental data[6] is analyzed[13,15]. The paradox is a consequence of heterogeneity due to the existence of more than one population of cells, which implies that in some fraction of cells, contact between two loci exists while in others it is absent. Each distinct population has its own statistics. For instance, the probability distribution of the spatial distance between the $m$th and the $n$th loci, $P_{i,mn}(r)$, for one population of cells could be different from another population of cells $P_{j,mn}(r)$, where $i$ and $j$ are the indices for the two different populations (Fig. 2a). The Hi-C experiments yield only an average value of the contact probability. Let us illustrate the consequence of the

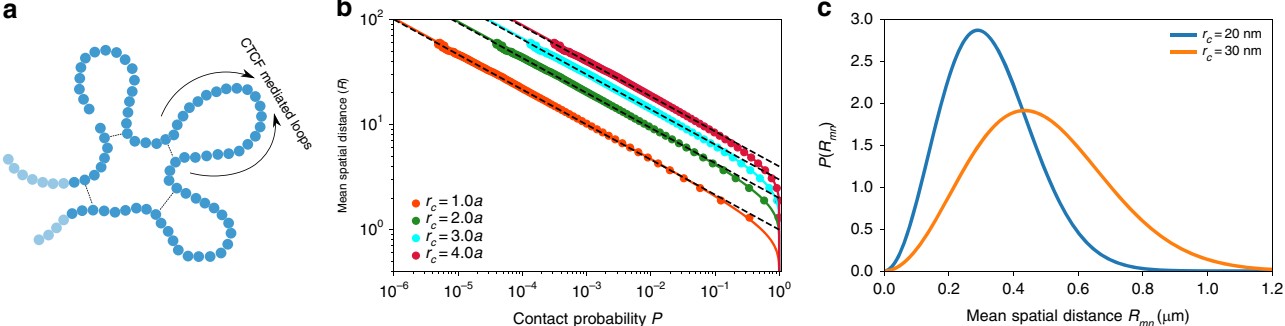

**Fig. 1** Simulations demonstrate the power-law relation between contact probability and mean spatial distance and the effect of $r_c$ on the inferred spatial distances. **a** A sketch of the Generalized Rouse Model for Chromosome (GRMC). Each bead represents a locus with a given resolution. Dashed lines represent harmonic bonds between loop anchors. **b** Mean spatial distance $\langle R \rangle$ as a function of the contact probability $P$. The solid lines are obtained using Eq. (1) for different values of $r_c$ (shown in the figure), the threshold distance for contact formation. The dots are simulation results. The agreement between simulations and theory is excellent. Asymptotically $\langle R \rangle$ approaches $r_c P^{-1/3}$ (dashed lines). The threshold for contact is expressed in terms of $a$, which is the equilibrium bond length in Eq. (15). **c** Illustration of the sensitivity of $r_c$ in determining the mean spatial distance $\langle R \rangle$. Blue and yellow curves are computed by solving $\langle R \rangle$ (Eq. (1)) for a given contact probability $P_{mn} = 10^{-3}$, and $r_c$. The calculated $\langle R_{mn} \rangle$ is used in Eq. (10) to obtain the distribution of the spatial distance $P(R_{mn})$. Blue and yellow curves are for the same value of $P$, but different $r_c$ values

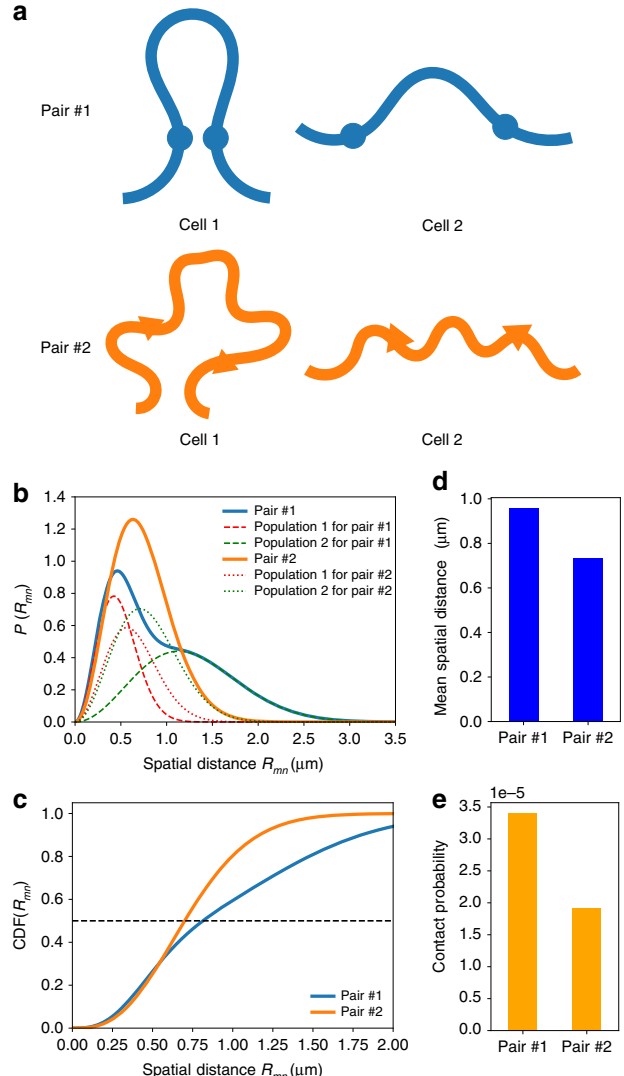

the other population. The probability distribution of spatial distance between the $m$th and the $n$th loci is a superposition of distributions for each population. Using Eq. (10), the mixed distribution can be written as,

$$P(R_{mn} = r) = \sqrt{\frac{2}{\pi}}\left( \eta\frac{r^2}{\sigma_{1,mn}^3}e^{-\frac{r^2}{2\sigma_{1,mn}^2}} + (1-\eta)\frac{r^2}{\sigma_{2,mn}^3}e^{-\frac{r^2}{2\sigma_{2,mn}^2}} \right),$$

(2)

where $\sigma_{1,mn}$ and $\sigma_{2,mn}$ are the parameters with different values characterizing the two populations. In the GRMC, $\sigma_{1,mn}$ and $\sigma_{2,mn}$ are related to the mean spatial distances in the two populations by $\langle R_{1,mn}\rangle = 2\sqrt{2/\pi}\sigma_{1,mn}$ and $\langle R_{2,mn}\rangle = 2\sqrt{2/\pi}\sigma_{2,mn}$. The mean spatial distance is, $\langle R_{mn}\rangle = \eta\langle R_{1,mn}\rangle + (1-\eta)\langle R_{2,mn}\rangle$, and the contact probability is $P_{mn} = \eta P_{1,mn} + (1-\eta)P_{2,mn}$, where $P_{1,mn}$ and $P_{2,mn}$ are the contact probabilities for each population, given by Eq. (1), which depends on the values of $\langle R_{1,mn}\rangle$ and $\langle R_{2,mn}\rangle$ as well as $r_c$.

If the values of $\langle R_{1,mn}\rangle$ and $\langle R_{2,mn}\rangle$ are unknown (as is the case in Hi-C experiments), and only the value of the contact probability between the two loci is provided, one can not uniquely determine the values of the mean spatial distances. This is the origin of the Hi-C and FISH data paradox. In Fig. 2b–e, we show an example of the paradox for a particular set of parameters ($\eta$, $\sigma_{1,mn}$, $\sigma_{2,mn}$). Pair #1 has a larger contact probability than pair #2, while also exhibiting a larger mean spatial distance. The GRMC explains in simple terms the origin of the paradox.

To systematically explore the parameter space, we display $\langle R_{mn}\rangle$ and $P_{mn}$ as heatmaps showing $\langle R\rangle_{1,mn}$ versus $\langle R\rangle_{2,mn}$ for different values of $\eta$ (Fig. 3). When there is a single homogenous population ($\eta = 0.0$), the mean spatial distance $\langle R_{mn}\rangle$ and contact probability $P_{mn}$ depend only on the value of $\langle R_{2,mn}\rangle$ (upper panel in Fig. 3). In this case, there is a precise one-to-one mapping between $\langle R_{mn}\rangle$ and $P_{mn}$. However, if $\eta \neq 0$ ($\eta = 0.3$, lower panel in Fig. 3) then the relation between $P_{mn}$ and $\langle R_{mn}\rangle$ is complicated. The contour lines for $P_{mn}$ cross the contour lines of $\langle R_{mn}\rangle$, which implies that for a given value of $P_{mn}$, one cannot infer the value of $\langle R_{mn}\rangle$ without knowing the value of $\eta$, $\langle R_{1,mn}\rangle$, and $\langle R_{2,mn}\rangle$. For instance, the triangle and circle shown for $\eta = 0.3$ in Fig. 3 demonstrate an example of the paradox, in which $\langle R(\blacktriangledown)\rangle (= 57a) > \langle R(\bullet)\rangle (= 40a)$ whereas $P(\blacktriangledown)(\approx 7.7\times 10^{-4}) > P(\bullet)(\approx 3.9\times 10^{-4})$.

**Extracting cell subpopulation information from FISH data.** Can we extract the information about subpopulations from experimental data so that the result from two vastly different techniques can be reconciled? To answer this question, we first generalize our theory for the GRMC to real chromatins. The generalization of Eq. (2) is,

$$P(R_{mn} = r) = \eta P(r|\langle R_{1,mn}\rangle) + (1-\eta)P(r|\langle R_{2,mn}\rangle),$$

(3)

where $P(r|\langle R_{1,mn}\rangle)$ and $P(r|\langle R_{2,mn}\rangle)$ are the Redner-des Cloizeaux distribution of distances for polymers[27,28] (Supplementary Note 1 and Supplementary Fig. 1). The distribution $P(r|\langle R_{mn}\rangle)$ is rigorously known for self-avoiding homopolymer in a good solvent, generalized Rouse model (Eq. (10) in the Methods section), and a semi-flexible polymer[29,30]. However, a simple analytic expression for chromosomes is not known. By assuming that the Redner-des Cloizeaux form for $P(r|\langle R_{mn}\rangle)$ also holds for chromosomes (see Supplementary Eq. (1) for details), we find that $g = 1$ and $\delta = 5/4$ in Supplementary Eq. (1). These parameters were previously extracted using the experimental data[8], and the Chromosome Copolymer Model (CCM) for chromosomes[24]. The value of $g$ is inferred from the scaling relationship between mean spatial distance $\langle R\rangle$ and contact probability $P$, $P \sim \langle R\rangle^{3+g}$. The value of $\delta$ is

**Fig. 2** Illustrating the FISH–Hi-C ($[P_{mn}, \langle R_{mn}\rangle]$) paradox. **a** Schematic illustration of the populations of two cells. There are two pairs of loci, pair 1 and pair 2. Cells 1 and 2 belong to two distinct populations such that pair 1 and pair 2 have different distributions of distances in the two cells. Pair 1 is always in proximity (contact is formed) in cell 1, whereas it is spatially separated (mean distance $>r_c$) in cell 2. Pair 2, on the other hand, has similar distributions of spatial distance in cells 1 and 2. Cell with two different populations gives rise to paradoxical behavior, which is illustrated by choosing $\eta_1 = 0.4$ and $\eta_2 = 1 - \eta_1 = 0.6$. These are the probabilities for a cell belonging to population 1 and 2, respectively. The pair 1 has parameters $\sigma_1 = 0.3\,\mu m$ and $\sigma_2 = 0.8\,\mu m$. The pair 2 has parameters $\sigma_1 = 0.4\,\mu m$ and $\sigma_2 = 0.5\,\mu m$. See Eq. (2) for the definition of $\sigma_1$ and $\sigma_2$. **b** The distribution of distance for pair 1 (thick blue) and pair 2 (thick orange), respectively. The distributions for the two different populations are shown separately for pair 1 (dashed lines) and pair 2 (dotted lines). **c** Cumulative distribution of the spatial distance. The horizontal dashed line indicates the median distance. **d** Mean distances for pair 1 is larger than for pair 2. **e** Pair 1 has larger contact probability than 2, which is paradoxical since the distance between the loci in pair 1 is larger than in 2. The threshold for determining contact is $r_c = 20\,nm$

inevitable heterogeneous mixture of cell populations by considering the simplest case in which only two distinct populations, one with probability $\eta$ and the other $1 - \eta$, are present (a generalization is presented below). For instance, in one population of cells, there is a CTCF loop between $m$ and $n$, and it is absent in

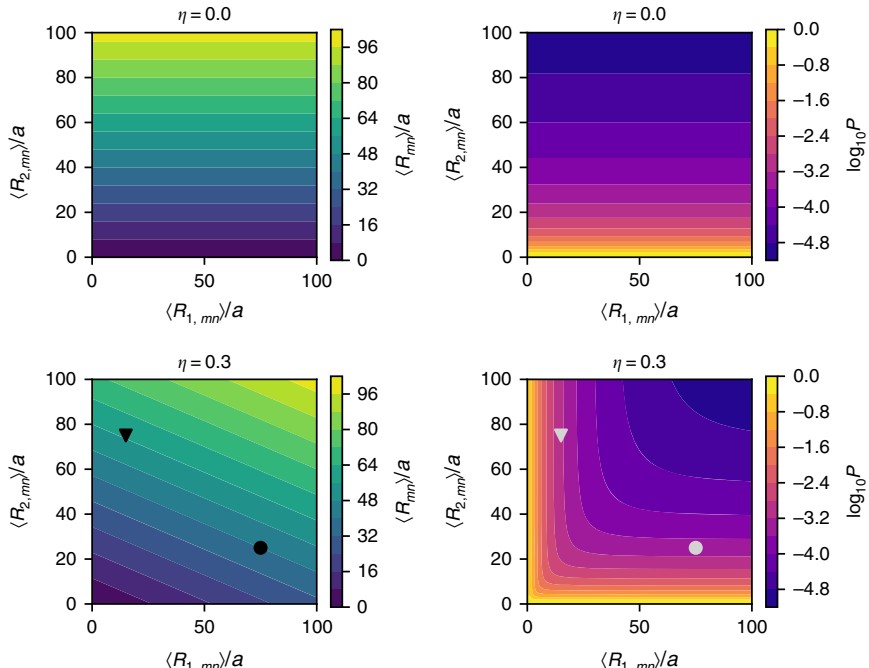

**Fig. 3** Plots of mean distance $\langle R_{mn} \rangle$ and the contact probability $P_{mn}$ as heatmaps computed using $r_c = 2a$. The colorbars on the right show the values of $\langle R_{mn} \rangle$ and $P_{mn}$. The results for $\eta = 0 (\neq 0)$ is shown on the top panel (bottom panel). Two specific pairs are marked as triangle and circle in the lower-left panel. These loci pairs illustrate the $[P_{mn}, \langle R_{mn} \rangle]$ paradox

computed as $\delta = 1/(1 - v)$. $v$ is inferred from scaling $\langle R(s) \rangle \sim s^v$, where $s$ is the genomic distance.

The integral of Eq. (3) up to $R$, which is the cumulative distribution function CDF($R$), can be used to fit the FISH data. Thus, the probability of contact formation can be computed as, $\int_0^{r_c} P(r|\langle R \rangle) dr$, where $r_c$ is the contact threshold. Using the data in ref. [6], the CDF($R$) for two pairs of loci are shown in Fig. 4a. By fitting the two experimentally measured curves to the theoretical prediction (see Supplementary Note 2), we obtain $\eta \approx 0.42$ for peak4-loop and $\eta \approx 0.97$ for peak3-control. The parameters obtained can then be used to compute the contact probability. Since the Hi-C experiments measure the number of contact events instead of contact probability and the value of $r_c$ is unknown, we compare the relative contact frequency, which is computed as $P_i/\langle P \rangle$, where $P_i$ is the contact probability computed using the model or the contact number measured in Hi-C for the $i$th pair and $\langle P \rangle$ is the mean value for all the pairs considered. First, we fit all the eight CDF($R$) curves in ref. [6]. the excellent agreement between theory and experiments is vividly illustrated in Supplementary Fig. 2 and also manifested by the Kolmogorov–Smirnov statistics (Supplementary Note 5 and Supplementary Table 1). Second, we calculate their corresponding relative contact frequency (Fig. 4b). Comparison of the theoretical calculations with Hi-C measurements shows excellent agreement (Fig. 4b) with the Pearson correlation coefficient being 0.87. The contact probability is computed using $r_c = 10$ nm. Note that any value of $r_c \leq 10$ nm gives similar results (Supplementary Fig. 3). The goodness of fits using different sets of $g$ and $\delta$ is summarized in Supplementary Table 2. The set of $g = 0$ and $\delta = 2$ gives equivalent good fits as the set of $g = 1$ and $\delta = 5/4$. It is also important to note that fitting the FISH data with the assumption that cell population is homogeneous leads to unphysical values of $g$ and $\delta$ and the Kolmogorov–Smirnov statistics are inferior (see Supplementary Note 4, Supplementary Fig. 5 and Supplementary Table 3).

Interestingly, the values of $\langle R_1 \rangle$ obtained from fitting the four CTCF/cohesin-mediated loops (peak(1, 2, 3, 4)-loop) are all about $0.25 - 0.35$ μm ($R_{1,\text{peak1-loop}} \approx 0.24$ μm, $R_{1,\text{peak2-loop}} \approx 0.33$ μm, $R_{1,\text{peak3-loop}} \approx 0.35$ μm, $R_{1,\text{peak4-loop}} \approx 0.30$ μm) regardless of their genomic separation (see Supplementary Table 1), suggesting that the mechanism of looping between CTCF motifs are similar with a mean spatial distance $\approx 0.3$ μm. The physically reasonable value of $\langle R_{mn} \rangle \approx 0.3$ vm for all peak–loop pairs shows that these CTCF-mediated contacts describe molecular interactions between loci that are separated by a few hundred kilo base pairs. It has been shown that these contacts, referred to as peaks[6] are significantly closer in space than others that are separated by similar genomic distance. The peak–loop contacts correspond to chromatin loops with the loci in the peaks being the anchor points between a specific loop. In sharp contrast, the distances between peak$i$ and control ($i$ goes from 1 to 4), which are greater than the distances between peak loci, vary ranging from $\approx 0.47$ to $\approx 0.67$ μm (see Supplementary Table 1). It is likely that these contacts are more dynamic because they are not be anchored by CTCF-binding proteins.

**Massive heterogeneity in chromosome organization**. In a recent study[19], which combined Hi-C and high-throughput optical imaging to map contacts within single chromosomes in human fibroblasts, revealed massive heterogeneity. Such extensive existence of a large number of conformations, leading to multiple or nearly continuous distribution of subpopulations, was much greater than previously anticipated. Although, the results in Fig. 4 quantitatively reveal heterogeneity associated with CTCF loops by considering only two dominant subpopulations, the most recent experiment requires a generalization of the theory. In principle, our theory also applies to interactions of any nature, not only the CTCF loops. In doing so, it may be more reasonable to assume a continuous distribution of subpopulations, $P(\langle R \rangle)$, (see Supplementary Notes 6 and 7, and Supplementary Fig. 6 for generalization) instead of two discrete subpopulations, $\langle R_1 \rangle$ and $\langle R_2 \rangle$, which of course is much simpler and may suffice in many cases as the results in Fig. 4 illustrate. As a proof of concept of our theory, we solve $P(\langle R \rangle)$ for the eight pairs of contacts analyzed in the

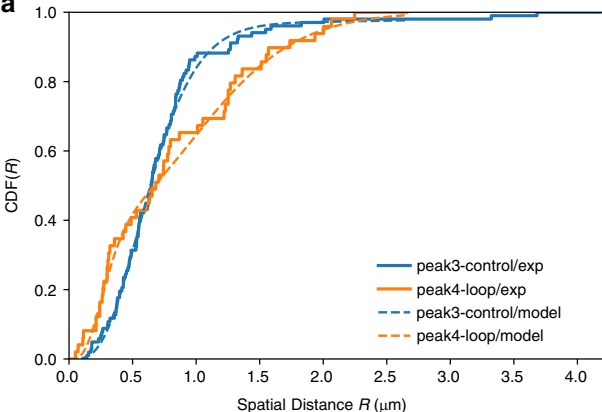

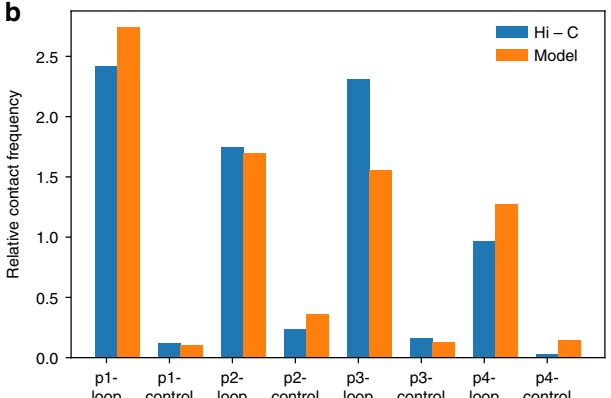

**Fig. 4** Extracting statistics of subpopulations from FISH data. **a** Cumulative distribution function of the spatial distance, CDF($R$) for two pairs of loci, labeled peak3-control and peak4-loop in ref. [6]. The excellent agreement between theory and experiments shows the usefulness of the relationship between $P_{mn}$ and $R_{mn}$ obtained using GRMC. The solid curves are the experiment data[6]. The dashed lines are the fits to $\int_0^R P(r)dr$ (the needed expressions are in Eq. (3) and Supplementary Eq. (1)). The best fit parameters are $\eta_{peak3-control} \approx 0.97$, $\langle R_{1,peak3-control}\rangle \approx 0.67$ μm, $\langle R_{2,peak3-control}\rangle \approx 4.08$ μm, $\eta_{peak4-loop} \approx 0.42$, $\langle R_{1,peak4-loop}\rangle \approx 0.30$ μm and $\langle R_{2,peak4-loop}\rangle \approx 1.21$ μm. **b** Relative contact frequency computed from the fits of CDF($R$) for eight pairs of loci investigate experimentally[6] (orange bars). For each pair of loci, the contact probability is calculated as $P_{mn} = \int_0^{r_c} P(r)dr$ (Eq. (3)) using the parameters obtained by fitting CDF($R$) with $r_c = 20$ nm. Comparison of the CDF($R$)s between theory and experiments for the eight pairs of loci are displayed in Supplementary Fig. 2. Blue bars are computed using the contact number from Hi-C measurements in ref. [6]. The relative contact frequency is calculated as $_{Pi}/\langle P\rangle$, where $_{Pi}$ is the contact probability computed using the model or the contact number measured in Hi-C for $i$th pair, and $\langle P\rangle$ is the mean value for all the pairs considered. p1-loop/p1-control/... are the ones referred to peak1-loop/peak1-control/... in ref. [6]

previous section. The results are shown in Supplementary Fig. 7. In all cases, $P(\langle R\rangle)$s are found to be multimodal. For peak1/2/3/4-control and peak3-loop, $P(\langle R\rangle)$ yield peaks located at positions very close to $\langle R_1\rangle$ and $\langle R_2\rangle$ shown in Supplementary Table 1, justifying the effectiveness of the theory. To show that our theory has a broader range of applicability, we use the FISH data from the recent study[19], which reports spatial distance measurements for 212 pairs of loci. $P(\langle R\rangle)$ is solved for each of a total of 212 pairs of loci. To illustrate our results, we compare in Fig. 5 the predicted CDF($r$) and the experimentally measured CDF($r$), as well as the $P(\langle R\rangle)$ obtained by fitting for six pairs of loci as examples in Fig. 5. The results show substantial variations in $\langle R\rangle$, manifested by the multiple peaks and wide spread variations in

$P(\langle R\rangle)$. Remarkably, the calculated CDF($r$) (without any adjustable parameters) and the measured CDF($r$) are in excellent agreement for the six loci pairs, which were arbitrarily chosen for illustration purposes. The residual errors between the two, shown as insets in Fig. 5, are extremely small.

In Fig. 6a, we show the normalized distributions $P(\langle R\rangle/\mu(\langle R\rangle))$ for each of the 212 pairs of loci (see Supplementary Fig. 8 for each pair as a separate figure). We expect that $P(\langle R\rangle/\mu(\langle R\rangle))$ should be narrowly distributed around value 1 if there is only one population. However, many $P(\langle R\rangle/\mu(\langle R\rangle))$ show multiple peaks with large variations. To further quantify the extent of heterogeneity, we calculate the coefficient of variation, $CV = \sigma(\langle R\rangle)/\mu(\langle R\rangle)$, where $\sigma(\langle R\rangle)$ and $\mu(\langle R\rangle)$ are the standard deviation and the mean of $\langle R\rangle$, respectively. If there is only one population associated with $\langle R\rangle$, CV should have a value of around zero. Figure 6b shows the histogram of CV for all 212 pairs of loci. The CV values are widely distributed, suggesting that 3D structural heterogeneity is common and is associated with many pairs of loci rather than a few. Thus, the analyses of experimental data are not possible without taking heterogeneity into account. The theory presented here is sufficiently general and simple that it can be used to calculate the measurable quantities readily.

**The role of loop extrusion in chromosome heterogeneity**. What is the origin of heterogeneity in the individual cell populations? There are two possibilities. The first one is static heterogeneity: each subpopulation explores a distinct region of the genomic folding landscape (GFL) (Fig. 7a). The second is the dynamic heterogeneity. Each cell explores a local minimum of the GFL before transiting to another local minimum (Fig. 7b). The only assumption in the application of our theory to genome organization is that there must be more than one population of cells, which does not violate the observation that the Hi-C experiment report only the average contact probability over millions of cells. Dynamic looping would be an example of the dynamic heterogeneity where the CTCF/cohein-mediated loops are formed and broken dynamically on a fast time scale compared with the lifetime of a cell. Such a picture is supported by recent single-cell molecule experiments[31,32]. The average residence time of CTCF/cohesin complex is shown to be in the range of a few to tens of minutes, which is much smaller compared with the time scale of the cell cycle (15–30 h). Loop extrusion model[33–35] is another possible origin of dynamic heterogeneity. In the loop extrusion model, it is thought that cohesins extrude loops along the chromosome fiber, which could detach stochastically. At any given time, there would be many subpopulations, each characterized by a distinct set of loops in the chromosome. Indeed, our analyses of the most recent high-throughput optical imaging data lend credence to the notion that multiple subpopulations in chromosomes arise because of massive dynamic heterogeneity. Our theory also gives an indirect theoretical justification for the work in ref. [15], in which the authors found the loop extrusion model could lead to the $[P_{mn}, \langle R_{mn}\rangle]$ paradox.

Single-cell temporal information is necessary to determine whether the loops are static or dynamic or a combination of the two (Fig. 7c). Hence, the combination of the dynamic FISH technique such as CRISPR–dCas9 FISH and single-cell Hi-C would be crucial for us to fully understand the organization of genomes. Our theory provides a theoretically rigorous method based on polymer physics to connect the results from measurements using the two vastly different techniques.

**Discussion**
From polymer physics for single chains it follows that in a homogeneous system, the contact probability and mean 3D

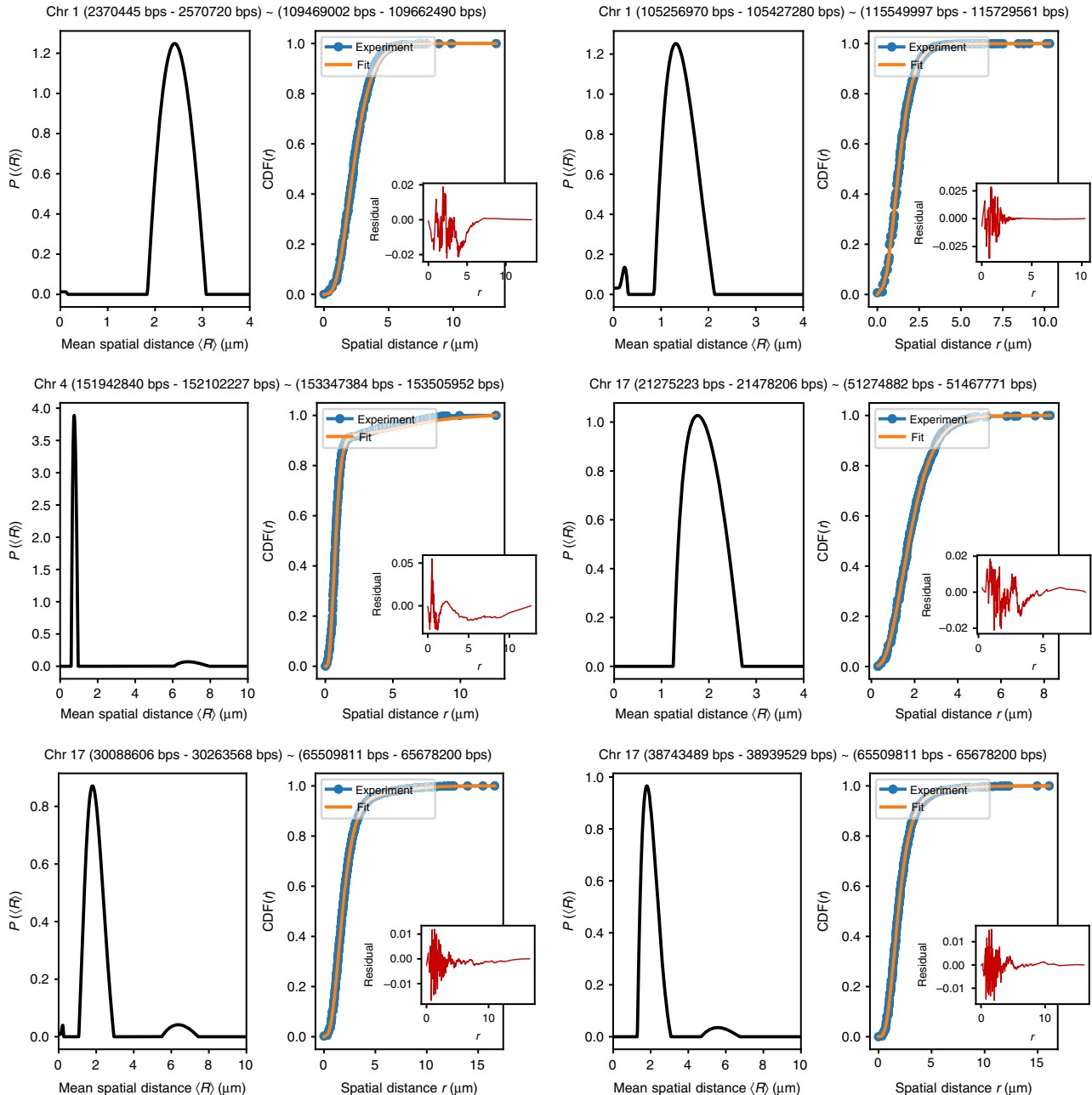

**Fig. 5** Exampled fits of CDF(r) using Supplementary Eq. (9) to the experimental data[19]. The six exampled pairs of loci are indicated above each subfigure. Orange lines, showing the fits using our theory, are indistinguishable from the experiment (the differences between fitted and experimental curve are shown in the insets). The distribution $P(\langle R \rangle)$ given in the integral equation (Supplementary Eq. (9)) is solved using nonnegative Tikhonov Regularization (Supplementary Note 7). As shown here, $P(\langle R \rangle)$ have multi-peaks and are widespread, which is a manifestation of heterogeneity. We set $g = 1$ and $\delta = 5/4$

distances are linked, resulting in a power-law relation connecting the two quantities that can be measured using Hi-C and FISH techniques. However, the one-to-one mapping does not hold in Hi-C experiments because of the presence of a mixture of distinct cell subpopulations each characterized by its own statistics leads to heterogeneity, which in turn gives rise to the $[P_{mn}, \langle R_{mn} \rangle]$ paradox. We show that the theory based on precisely solvable GRMC could be used to solve the paradox in practice. The theory can be readily used to analyze data from experiments, provided the FISH and Hi-C experiments are done under similar conditions[6]. The central result of the theory in Eq. (3) can be used to analyze the available sparse FISH data. We show that the fraction of cell subpopulations ($\eta$ in Eq. (3)) and the generalization derived in Supplementary Note 6 can be extracted by fitting the

FISH data using our theory. From Eq. (3), we calculate the Hi-C contact probabilities, thus establishing that the theory resolves the $[P_{mn}, \langle R_{mn} \rangle]$ paradox.

In this work, we confine ourselves to two-point interactions, which allows us to consider one pair of loci at a time. However, recent experiments probing multipoint interactions have suggested that formations of loops are likely to be cooperative[9,36], such that the formation of one loop could facilitate the formation of a nearby loop. Such cooperative loop formation was previously shown in an entirely different context involving the folding of proteins directed by disulfide bond formation[37]. It can be shown within our framework that the formation of one loop can certainly increase the probability of formation of another loop. The theoretical basis for this finding is given in the Supplementary Note 8.

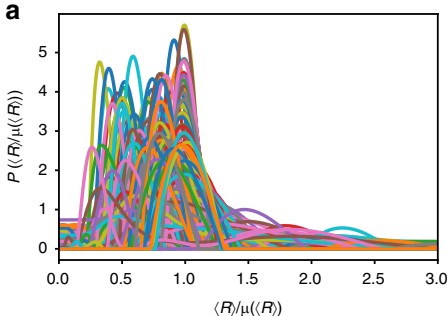
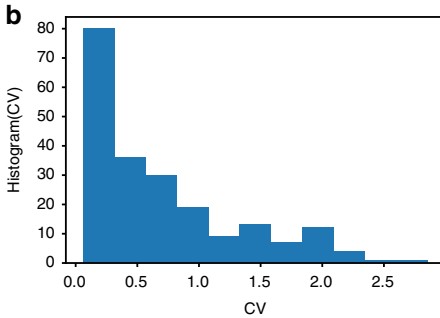

**Fig. 6** Chromosome conformations are extensively heterogeneous. **a** Normalized distribution $P(\langle R\rangle/\mu(\langle R\rangle))$ ($\mu(\langle R\rangle)$ is the mean of $\langle R\rangle$) for all the 212 pairs of loci reported in ref. [19]. For almost every pair of loci, the associated $P(\langle R\rangle/\mu(\langle R\rangle))$ has multiple peaks, and is widespread. **b** Histogram of the coefficient of variations CV for all 212 pairs of loci probed in ref. [19]. The CV values are calculated for each pair of loci, using CV $= \sigma(\langle R\rangle)/\mu(\langle R\rangle)$, where $\sigma(\langle R\rangle)$ is the standard deviation of $\langle R\rangle$. For a large number of loci pairs, CV exceeds 0.5, which is a quantitative measure of the extensive heterogeneity noted in the experiment[19]

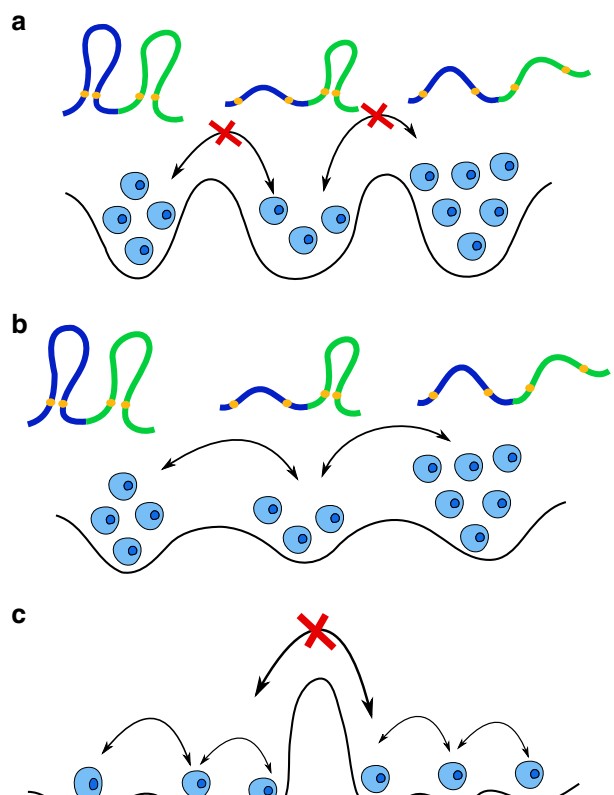

**Fig. 7** Schematic of the Genomic Folding Landscape (GFL). **a** Static heterogeneity: cell subpopulation occupies distinct local minima in the GFL, with each minimum representing a stable organization. The energy barrier is too large for transitions between different local minima on a biological time scale (one cell cycle). **b** Dynamical heterogeneity: the energy barrier between local minima on the landscape is small enough which allows the dynamic transition between different subpopulations. **c** Combination of two different types of heterogeneity. In all three scenarios, the $[P_{mn}, \langle R_{mn}\rangle]$ paradox arises. The loci contacts are in orange. The polymer conformation sketches are not shown in this scenario due to insufficient space

The reconciliation of the FISH and Hi-C data using polymer physics concepts is the first key step in integrating the data from these experimental techniques to construct the 3D structures of chromosomes. The work described here provides a theoretical basis for accomplishing this important task. Finally, our results suggest that heterogeneity in contact formation is an intrinsic property of genome organization, and hence the acquisition of single-cell experimental data is crucial to further our understanding of both the dynamics and the heterogeneous structural organization of chromosomes.

## Methods

**Generalized rouse model for chromosome**. In order to derive an approximate relationship connecting contact probabilities between loci and the three-dimensional distances, we use a variant of the random loop model[38,39]. We first consider a minimal cross-linked phantom chain model, which incorporates the presence of CTCF/cohein-mediated loops[6]. The model, originally introduced for describing physical gels[38], and more recently used for chromosome dynamics in a number of insightful studies[23,39], could be viewed as a Generalized Rouse Chromosome Model (GRMC)[40,41]. The cross-links modeling the CTCF/cohein-mediated loops here are not random. Their locations are predetermined by the Hi-C data[6].

The equation of motion for the GRMC is[42]

$$\xi\frac{d\mathbf{R}}{dt} = \mathbf{AR} + \mathbf{F}, \tag{4}$$

where $\xi$ is the friction coefficient, $\mathbf{R} = [\mathbf{r}_1, \mathbf{r}_2, ..., \mathbf{r}_N]^T$ with $\mathbf{r}_i$ being the position of the $i$th locus. The vector $\mathbf{F} = [\mathbf{f}_1, \mathbf{f}_2, ..., \mathbf{f}_N]^T$ (T is the transpose), where $\mathbf{f}_i$ is the Gaussian random force acting on the $i$th locus, characterized by $\langle f_n(t)\rangle = 0$ and $\langle f_{n\alpha}(t)f_{m\beta}(t')\rangle = 2\xi k_B T\delta_{nm}\delta_{\alpha\beta}\delta(t - t')$; $\mathbf{A}$ is the $N\times N$ connectivity matrix, embedding the information of chain connectivity and the location of the loops connecting two loci (Fig. 1a)

$$A_{mn} = \begin{cases} -2\kappa - |\Sigma_m|\omega, & \text{if } m = n\neq1 \text{ or } N \\ -\kappa - |\Sigma_m|\omega, & \text{if } m = n = 1 \text{ or } N \\ \kappa, & \text{if } |m - n| = 1 \\ \omega, & \text{if } |m - n|>1, \text{ and connected in } \Sigma \\ 0, & \text{if otherwise} \end{cases} \tag{5}$$

where $\Sigma$ is the set of indices representing the loci pairs specifying the CTCF facilitated loop anchors, and $|\Sigma_m|$ is the number of loops connected to the $m$th locus. The spring constant $\kappa$ enforces chain connectivity, and $\omega$ is the associated spring constant for a CTCF pair. Note that the GRMC model does not account for excluded volume interactions, which in the modeling of chromatin is often justified by noting that topoisomerases enable chain crossing. Our purpose is to use GRMC to first illustrate concretely the challenges in going from the measured average contact map to spatial organization, precisely. More importantly, using the insights from the study of the GRMC, we provide a solution to the FISH–Hi-C paradox.

Since $\mathbf{A}$ in Eq. (5) is a real symmetric matrix, it can be diagonalized using the orthonormal matrix $\mathbf{V}$

$$\mathbf{VAV}^T = \mathbf{\Lambda} = \text{diag}(\lambda_0, \lambda_1, ..., \lambda_{N-1}), \tag{6}$$

where $\lambda_0, \lambda_1, ..., \lambda_{N-1}$ are the eigenvalues of $\mathbf{A}$. By defining $\mathbf{X} = \mathbf{VR}$ and using $\mathbf{R} = \mathbf{V}^T\mathbf{X}$ and $\mathbf{VV}^T = \mathbf{I}$, we obtain the equations of motion of the normal coordinates $\mathbf{X}$

$$\xi\frac{d\mathbf{X}}{dt} = \mathbf{\Lambda X} + \mathbf{f}. \tag{7}$$

Because $\mathbf{\Lambda}$ is a diagonal matrix, the normal coordinates of the GRMC $\mathbf{X}_p$ are decoupled. Using the normal modes, $\mathbf{X}$, the physical quantities associated with the polymer can be readily calculated. Therefore, for GRMC with a predetermined set of CTCF/cohein-mediated loops, we can solve for the eigenvalues of the connectivity matrix $\mathbf{A}$, and the orthonormal matrix $\mathbf{V}$ numerically, and thus calculate the contact probability and spatial distance precisely.

**Relation between contact probability and mean spatial distance**. The vector between the positions of the $m$th and the $n$th loci may be written as

$$\mathbf{R}_m - \mathbf{R}_n = \sum_{p=0}^{N-1} (V_{pm} - V_{pn}) \mathbf{X}_p, \tag{8}$$

where $V_{pm}$ and $V_{pn}$ are the elements of orthonormal matrix $\mathbf{V}$. The equilibrium solution of Eq. (7) yields, $\lim_{t\to\infty} X_{p,\alpha}(t) \sim \mathcal{N}(0, -\frac{k_\mathrm{B}T}{\lambda_p})$, where $\alpha = x, y, z$, $\mathcal{N}$ is Gaussian distribution. Therefore

$$\lim_{t\to\infty} R_{mn,\alpha}(t) \sim \mathcal{N}\left(0, -\sum_{p=0}^{N-1} (V_{pm} - V_{pn})^2 \frac{k_\mathrm{B}T}{\lambda_p}\right) \equiv \mathcal{N}(0, \sigma_{mn,\alpha}^2). \tag{9}$$

where $\sigma_{mn,\alpha} = -\sum_{p=0}^{N-1} (V_{pm} - V_{pn})^2 (k_\mathrm{B}T/\lambda_p)$. Since the model is isotropic, it follows that $\sigma_{mn,x}^2 = \sigma_{mn,y}^2 = \sigma_{mn,z}^2 \equiv \sigma_{mn}^2$. The mean distance $\langle R_{mn} \rangle$ is related to $\sigma_{mn}$ through $\langle R_{mn} \rangle = 2\sqrt{2/\pi}\sigma_{mn}$. The distribution of the distance between the $m$th and the $n$th loci, $\lim_{t\to\infty} |\mathbf{R}_{mn}(t)| = \lim_{t\to\infty} \sqrt{\sum_\alpha R_{mn,\alpha}^2(t)}$ is a non-central chi distribution (we will neglect the notation $\lim_{t\to\infty}$ from now on)

$$P(R_{mn} = r) = \sqrt{\frac{2}{\pi}} \frac{1}{\sigma_{mn}} e^{-r^2/(2\sigma_{mn}^2)} \frac{r^2}{\sigma_{mn}^2}. \tag{10}$$

The contact probability $P_{mn}$, for a given threshold $r_\mathrm{c}$ (contact exists if $r \le r_\mathrm{c}$), computed using Eq. (10) yields

$$\begin{aligned} P_{mn} &= \int_0^{r_\mathrm{c}} \mathrm{d}r \sqrt{\frac{2}{\pi}} \frac{1}{\sigma_{mn}} e^{-r^2/(2\sigma_{mn}^2)} \frac{r^2}{\sigma_{mn}^2} \\ &= \mathrm{Erf}\left(\frac{r_\mathrm{c}}{\sqrt{2}\sigma_{mn}}\right) - \sqrt{\frac{2}{\pi}} e^{-\frac{r_\mathrm{c}^2}{2\sigma_{mn}^2}} \frac{r_\mathrm{c}}{\sigma_{mn}}. \end{aligned} \tag{11}$$

The mean spatial distance $\langle R_{mn} \rangle$ is given by

$$\langle R_{mn} \rangle = \int_0^\infty \mathrm{d}r\, r \sqrt{\frac{2}{\pi}} \frac{1}{\sigma_{mn}} e^{-r^2/(2\sigma_{mn}^2)} \frac{r^2}{\sigma_{mn}^2} = 2\sqrt{\frac{2}{\pi}}\sigma_{mn}. \tag{12}$$

Using Eqs. (11) and (12), the desired relation between $P_{mn}$ and $\langle R_{mn} \rangle$ becomes

$$P_{mn} = \mathrm{erf}\left(\frac{2r_\mathrm{c}}{\sqrt{\pi}\langle R_{mn}\rangle}\right) - \frac{4}{\pi} \frac{r_\mathrm{c}}{\langle R_{mn}\rangle} e^{-\frac{4r_\mathrm{c}^2}{\pi\langle R_{mn}\rangle^2}} \equiv R_0(\langle R_{mn}\rangle). \tag{13}$$

Eq. (13) is identical with Eq. (1) in the main text.

**Simulations**. The energy function for the GRMC is

$$U(\mathbf{r}_1, ..., \mathbf{r}_N) = \sum_{i=1}^{N-1} U_i^\mathrm{S} + \sum_{\{p,q\}} U_{\{p,q\}}^\mathrm{L}. \tag{14}$$

For the bonded stretch potential, $U_i^\mathrm{S}$, we use

$$U_i^\mathrm{S} = \frac{\kappa}{2} (|\mathbf{r}_{i+1} - \mathbf{r}_i| - a)^2, \tag{15}$$

where $a$ is the equilibrium bond length. The interaction between the loop anchors is also modeled using a harmonic potential

$$U_{\{p,q\}}^\mathrm{L} = \frac{\omega}{2} (|\mathbf{r}_p - \mathbf{r}_q| - a)^2, \tag{16}$$

where the spring constant is associated with the CTCF facilitated loops, and $\{p, q\}$ represents the indices of the loop anchors, which are taken from the Hi-C data[6] (Supplementary Note 3). We simulate the chromosome segment from 146 to 158 Mbps of Chromosome 5. Each monomer represents 1200 bps, resulting the total number of coarse-grained loci $N = 10,000$.

In order to accelerate conformational sampling, we perform Langevin Dynamics simulations at low friction[43]. We simulate each trajectory for $10^8$ time steps, and save the snapshots every 10,000 time steps. We generate ten independent trajectories, which are sufficient to obtain reliable statistics (Supplementary Fig. 4).

**Data analyses**. The contact probability between the $m$th and $n$th loci in the simulation is calculated using

$$P_{mn} = \frac{1}{TM} \sum_{a=1}^{M} \sum_{t=1}^{T} \Theta(r_\mathrm{c} - |\mathbf{r}_m^{(a)}(t) - \mathbf{r}_n^{(a)}(t)|), \tag{17}$$

where $\Theta(\cdot)$ is the Heaviside step function, $r_\mathrm{c}$ is the threshold distance for determining the contacts, the summation is over the snapshots along the trajectory, and the total $M$ number of independent trajectories, and $T$ is the number of snapshots for a single trajectory. The mean spatial distance between the $i$th and the $j$th loci in the simulation is calculated using

$$\langle R_{mn} \rangle = \frac{1}{TM} \sum_{a=1}^{M} \sum_{t=1}^{T} |\mathbf{r}_m^{(a)}(t) - \mathbf{r}_n^{(a)}(t)|. \tag{18}$$

The objective is to go from $P_{mn}$ to $\langle R_{mn} \rangle$, and to determine, if in doing so, we get reasonable results. Because these quantities can be computed precisely in the GRMC, the $[P_{mn}, \langle R_{mn} \rangle]$ relationship can be tested, which allows us to obtain the needed cues to solve the FISH–Hi-C paradox.

**Reporting summary**. Further information on research design is available in the Nature Research Reporting Summary linked to this article.

## Data availability
All relevant data supporting the findings of this study are available within the article and its Supplementary Information files or upon requests from the corresponding author. The Hi-C and FISH experimental data used in this study are publicly available from GEO database under accession number GSE63525 and from 4DN portal at https://data.4dnucleome.org/publications/80007b23-7748-4492-9e49-c38400acbe60/. The processed data are available upon request from the authors.

## Code availability
The polymer simulations are performed using LAMMPS Molecular Dynamics Simulation software[44], which is an open-source code available at http://lammps.sandia.gov. The codes used to analyze data in the present study are deposited to Github repository https://github.com/anyuzx/chromosome-heterogeneity-analysis.

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

## Acknowledgements

We are grateful to the National Science Foundation (CHE 19-00093) and the Collie-Welch Regents Chair (F-0019) for supporting this work.

## Author contributions

G.S. and D.T. designed and performed the research, G.S. and D.T. analyzed the data, G.S. and D.T. wrote the paper.

## Additional information

**Competing interests:** The authors declare no competing interests.

