## [Peer Review File · Nature Communications]

Reviewers' comments:

Reviewer #1 (Remarks to the Author):

Understanding the measurements of Hi-C and FISH within a consistent model is of great importance, particularly as both of these technologies are being employed in more and more biological investigations, sometimes with conflicting results.

In this study the authors develop upon a specific polymer model (the Generalized Rouse Model for Chromosomes) to incorporate the heterogeneity of different subpopulation of cells. More specifically, they employ a mixture model to extract the probability of CTCF loop formation within the population. While minimalist, the model provides analytical solutions that were used to readily fit available two-color FISH data. Within this framework the divergence between Hi-C and FISH measurements readily follows from cell heterogeneity.

Before publication I believe the authors should address the following points:

- 1) In recent years there have been numerous experimental studies addressing the heterogeneity of chromatin structure across cells with both Hi-C, particularly with the advent of single-cell Hi-C, (Stevens et al Nature 2017, Flyamer et al Nature 2017, Tan et al Science 2018 etc.) and with multiplexed FISH (Wang et al Science 2016, Bintu et al Science 2018 etc.). First the current state of the field should be properly introduced. Second, all these studies provide available online data. The author's model should be stressed with more experimental data.
- 2) The FISH-Hi-C paradox, while a provocative name, should be introduced more appropriately otherwise it is misleading to the readers. For the most part, FISH and Hi-C agree remarkably well (see Wang et al Science 2016, Bintu et al Science 2018, and recent bioarxiv works from Misteli and Cavalli lab).
- 3) This study focuses on a very specific chromatin feature - CTCF loop formation. However, this is expected to form in a more complicated context where multivalent protein interactions can give rise to domains and compartments. Can the authors provide more insights on how their model can be modified to incorporate these effects? Are analytical extensions possible and what insight can be gained in understanding heterogeneity by employing such models?
- 4) Numerous recent experimental studies aim to measure multi-point interaction with technologies such as TriC, Sprite, multiplexed FISH etc. These studies suggest that loops work cooperatively such that the formation of a loop can facilitate the formation of a neighboring loop (possibly through the collision of loop extruders). Can the authors comment on how this phenomenon can be modeled within their framework?

Reviewer #2 (Remarks to the Author):

This manuscript proposes a theoretical framework based on polymer physics and a generalised Rouse model to explain previously reported discrepancies between Hi-C data measuring contact frequencies between any two points on a chromosome and FISH experiments giving the distribution of distances between two probes in a chromosome. A typical example of the apparent discrepancy is if the contact probability between a pair of locus is larger than that of another, but the distance of the second locus is on average smaller. The authors convincingly show that one possible simple mechanism to explain this apparent paradox (the Hi-C FISH paradox) is if the configurations sampled by the polymer are heterogeneous, so that there are two subpopulations of conformations (where the first pair for instance is either looped or not).

I think a theoretical contribution clarifying the relation between HiC and FISH is timely and welcome because these are two main methods for looking at chromosome structure. The idea that apparent discrepancies appear due to inhomogeneous populations of conformations (non mean-field/uniform populations) is reasonable and makes a lot of sense. I would say that it is not fully new as it is well recognised that the main difference between the methods is that Hi-C is a population averaged method whereas FISH gives the whole distribution of distances in the population (i.e., we compare an average in Hi-C with a full distribution in FISH). Still, the simplicity and generality of the current mathematical treatment are appealing hence I think this work is in principle suitable for Nat. Comm. I also have the following (minor) comments though:

1. There are quite a few papers comparing simulations with chromosome conformation capture methods (CaptureC or HiC) and simultaneously FISH, and it would be good to discuss these a bit more in detail. One example which is briefly discussed is Ref. 43, where the polymer model is fully compatible with both 5C and FISH. There are more recent ones as well, for instance in A. Buckle et al., Mol. Cell 72, 786 (2018), CaptureC and FISH are combined to give a suitable model for locus folding in mouse, and the large structural variability observed in conformations is relevant I believe for the heterogeneity in population assumed in the current work.

2. A key interesting contribution of the paper is the fit of FISH data from Rao et al to get the fraction of populations with or without loops (Fig. 4). However I'd say it is typically difficult to discriminate between a FISH distribution which is uniform and one which has multiple components/subpopulations. In the current case the fit is only possible as far as I can see because there are only two assumed subpopulations. What would happen if there are additional loops within the loop anchors (nested loops, or combination of nested and consecutive loops)? there I would expect the combinatorial possibilities for each loop would need to be considered rendering fitting impossible/not too meaningful. In other words this method is mainly usable only for relatively small

CTCF loops which do not have further loops nested in their interior, and I am not sure whether this is the rule or the exception in the data. Can the authors comment more on this and on how this discussion would change if more than two populations are relevant?

3. In the discussion the authors mention dynamic loop extrusion as a possibility to lead to a heterogeneous population but this would be multiple rather than two subpopulations, right? Also, they correctly point to [24,25] to say that cohesin/CTCF has finite lifetimes on DNA. While cohesin residence time is 20 min (approx) the work of Ref. [25] suggests CTCF gets off in a time of order 1 minute, over which it is difficult to believe that extrusion can processively lead to loops of 100 kbp or more. Therefore I am not sure that citing [25] is a convincing argument to say that dynamic loop extrusion can be used to explain the HiC FISH paradox/chromosome conformation heterogeneity. If the authors agree I suggest they rephrase this next-to-last paragraph before the Conclusions sections.

4. On page 7 a reference appears as ? to the chromosome copolymer model. Also, can the authors give a bit more details here?

Reviewer #3 (Remarks to the Author):

Summary:

This paper designs a new Generalized Rouse Chromosome Model to explain the discrepancy between the chromosomal distance measured by FISH data and the chromosomal contact probability derived from the Hi-C data. It rigorously formulates the mathematical relationship between the distance and contact probability of loci and convincingly proves that the heterogeneity in the cell population is the factor causing the discrepancy. And it also presents an approach to calculate the fraction of sub cell populations from data. The theory agrees the experimental data well. The theory is successfully applied to the CTCF-mediated chromatin loops to calculate their distance. Overall, the theory is innovative and sound and the reasoning is logical. The theory is very useful for deriving the distance information from Hi-C and FISH data in order to build 3D structures of chromosomes and genomes.

Minor comments:

(1) The citation in this sentence “Chromosome Copolymer Model (CCM) for chromosomes [?]” is missing.

Responses to Reviewers:

Reviewer #1 (Remarks to the Author):

Understanding the measurements of Hi-C and FISH within a consistent model is of great importance, particularly as both of these technologies are being employed in more and more biological investigations, sometimes with conflicting results. In this study the authors develop upon a specific polymer model (the Generalized Rouse Model for Chromosomes) to incorporate the heterogeneity of different subpopulation of cells. More specifically, they employ a mixture model to extract the probability of CTCF loop formation within the population. While minimalist, the model provides analytical solutions that were used to readily fit available two-color FISH data. Within this framework the divergence between Hi-C and FISH measurements readily follows from cell heterogeneity.

Before publication I believe the authors should address the following points:

Response: The reviewer has succinctly summarized the major findings in the paper. Indeed, it is the simplicity of the theory that provided quantitative insight allowing us to arrive at the conclusions about the high degree of heterogeneity of chromosome organization. In what follows, we address the issues raised here.

1) In recent years there have been numerous experimental studies addressing the heterogeneity of chromatin structure across cells with both Hi-C, particularly with the advent of single-cell Hi-C, (Stevens et al Nature 2017, Flyamer et al Nature 2017, Tan et al Science 2018 etc.) and with multiplexed FISH (Wang et al Science 2016, Bintu et al Science 2018 etc.). First the current state of the field should be properly introduced. Second, all these studies provide available online data. The author's model should be stressed with more experimental data.

Response: We would like to thank the Reviewer for pointing out these references. We agree that the explanation that heterogeneity is the cause of the apparent paradox has considerable experimental support, as the Reviewer says. In the revised version we have added a discussion citing these important papers. At the time of writing of our paper we were not aware of the 2018 papers, unfortunately. It now reads on page 4 in the main text,

"By building on the GRMC results, we show that heterogeneity is the dominant feature of chromosome organization. Indeed, recent single-cell Hi-C [20–22] and FISH experiments [8, 9, 18, 19] have revealed that there are substantial cell-to-cell variations on genome organization. However, how to utilize the data reported in these experiments to enhance our understanding of 3D genome structural heterogeneity has not been unexplored. One approach is to create an appropriate polymer model based on Hi-C and imaging data,

which would readily allow us to probe the structural variability using simulations [23–26]. Indeed, it has been shown, using Hi-C and FISH data as well simulations [26], that if the conformation of the chromatin fiber is taken to be homogeneous then trends observed in the FISH data could not be predicted. However, using simulations and including two levels chromatin organization (open and compact) qualitative trends observed in the FISH data could be recovered [26].

Here, we develop a new and fully theoretical approach, which allows us to provide quantitative insights into the extent of heterogeneity in chromosome organization. From our theory it follows that the resolution of the FISH-Hi-C paradox requires invoking the notion of heterogeneity, which implies multiple populations of chromosomes coexist.”

In order to assess the concordance or discordance between Hi-C and FISH data it is necessary to have both sets of data. Moreover, in almost no case do we find the data in a tabular or easy-to-use form. Hence, we restricted ourselves to the data used in the Ref. 13, which prompted our study.

We did find additional data in the most recent paper by Misteli (cited as Ref. 19). The quantitative analyses of the data, using a generalization of the two-population model used to analyze Rao data, are presented in Figs. 5 (for 6 loci pairs) and 6 in the main text. The results for the all 212 pairs are given in the Supplementary Fig.8. The needed theory is described in the Supplementary notes 6 and 8.

2) The FISH-Hi-C paradox, while a provocative name, should be introduced more appropriately otherwise it is misleading to the readers. For the most part, FISH and Hi-C agree remarkably well (see Wang et al Science 2016, Bintu et al Science 2018, and recent bioarxiv works from Misteli and Cavalli lab).

Response: This is a very good point. We used the term paradox following the description in ref. 15 (Fudenberg and Imakaev). Of course, from a purist perspective even a single violation should be treated as counter-intuitive, which has to be explained. Having said this, we have made clear right at the outset that in most cases there is no qualitative discordance between Hi-C and FISH experiments. We ought to emphasize the extent of violation is not fully known because there is paucity of FISH data. Hence, it might be the case that there are more violations than is currently known. On page 2-3 in the main text, it now reads,

“It should be noted that in most cases, the Hi-C and FISH measurements agree very well [8, 9, 18, 19]. However from a purely theoretical perspective even a single contradiction is intriguing if the experimental errors can be ruled out. An outcome of our theory is that the discordance between FISH and Hi-C data arises because of extensive heterogeneity,

which is embodied by the presence of a variety of conformations adopted by chromosomes in each cell."

3) This study focuses on a very specific chromatin feature - CTCF loop formation. However, this is expected to form in a more complicated context where multivalent protein interactions can give rise to domains and compartments. Can the authors provide more insights on how their model can be modified to incorporate these effects? Are analytical extensions possible and what insight can be gained in understanding heterogeneity by employing such models?

Response: The reviewer is correct that there are may be multivalent interactions that produce the organization in chromosomes. In addition, these interactions could form in a cooperative fashion. The simplicity of model would allow us to extend it to include these effects. In the revised main text as well as newly added supplementary note 6 , we show how to extend our theory to FISH data for non-CTCF loop interactions. We use the data from recent work from Misteli's lab (a paper that was published only in March 2019) since it provides thousands of measurements for a large number of pairs of loci. We demonstrate that our method also works equally well for non-CTCF interactions. Our analyses of Misteli's data again show that there are substantial variations associated with not only short range but also long-range interactions.

On page 9-10 in the revised main text, we added a new section "Accounting for massive heterogeneity in chromosome organization". In this section, we analyzed the data for the 212 loci pairs using our theory with **no adjustable parameters**. In the main text we show the results in Fig. 5 (revised version) for six loci pairs. The agreement between the theoretical predictions and experiments is excellent. Similar results for the other loci pairs are given in the Supplementary Fig.8.

In order to obtain these results, we further developed the theory, which is described in Supplementary Note 6. The numerical procedure for solving the resulting integral equation is given in Supplementary Note 8.

4) Numerous recent experimental studies aim to measure multi-point interaction with technologies such as TriC, Sprite, multiplexed FISH etc. These studies suggest that loops work cooperatively such that the formation of a loop can facilitate the formation of a neighboring loop (possibly through the collision of loop extruders). Can the authors comment on how this phenomenon can be modeled within their framework?

Response: In part we remarked on it above. As the Reviewer appreciates, even the formation of a single loop is "many body" problem because of the polymeric nature of chromosomes. Such cooperative and possibly non-cooperative effects might manifest themselves in loop-loop interactions. If we are only concerned about the equilibrium aspects, then the correlation between two loops can be probed using the GRMC theory. We wrote the Supplementary Note 7 describing the theory for correlated loops.

In the revised main text, we also added a discussion of the cooperative interaction between multiple loops. On page 12 in the revised main text, it now reads,

"In this work, we confined ourselves to two-point interactions, which allows us to consider one pair of loci at a time. However, recent experiments probing multi-point interactions have suggested that formations of loops are likely to be cooperative [9, 37], such that the formation of one loop could facilitate the formation of a nearby loop. Such cooperative loop formation was previously shown in an entirely different context involving folding of proteins directed by disulfide bond formation [38]. It can be shown within our framework that the formation of one loop can certainly increase the probability of formation of another loop. The theoretical basis for this finding is given in the Supplementary Information Note 7."

Reviewer #2 (Remarks to the Author):

This manuscript proposes a theoretical framework based on polymer physics and a generalised Rouse model to explain previously reported discrepancies between Hi-C data measuring contact frequencies between any two points on a chromosome and FISH experiments giving the distribution of distances between two probes in a chromosome. A typical example of the apparent discrepancy is if the contact probability between a pair of locus is larger than that of another, but the distance of the second locus is on average smaller. The authors convincingly show that one possible simple mechanism to explain this apparent paradox (the Hi-C FISH paradox) is if the configurations sampled by the polymer are heterogeneous, so that there are two subpopulations of conformations (where the first pair for instance is either looped or not).

I think a theoretical contribution clarifying the relation between HiC and FISH is timely and welcome because these are two main methods for looking at chromosome structure. The idea that apparent discrepancies appear due to inhomogeneous populations of conformations (non mean-field/uniform populations) is reasonable and makes a lot of sense. I would say that it is not fully new as it is well recognised that the main difference between the methods is that Hi-C is a population averaged method whereas FISH gives the whole distribution of distances in the population (i.e., we compare an average in Hi-C with a full distribution in FISH). Still, the simplicity and generality of the current mathematical treatment are appealing hence I think this work is in principle suitable for Nat. Comm. I also have the following (minor) comments though:

Response: We are very pleased that this Reviewer appreciates our work, and states that the work is suitable for publication in this journal.

1. There are quite a few papers comparing simulations with chromosome conformation capture methods (CaptureC or HIC) and simultaneously FISH, and it would be good to

discuss these a bit more in detail. One example which is briefly discussed is Ref. 43, where the polymer model is fully compatible with both 5C and FISH. There are more recent ones as well, for instance in A. Buckle et al., Mol. Cell 72, 786 (2018), CaptureC and FISH are combined to give a suitable model for locus folding in mouse, and the large structural variability observed in conformations is relevant I believe for the heterogeneity in population assumed in the current work.

Response: We thank the Reviewer for pointing the paper by A. Buckle. We have cited it in the current version and provided a brief discussion of how it does tie in with the notion of heterogeneity shown to be necessary for quantitatively reproducing the data in the original version and the new theoretical predictions made in the revised version in an attempt to explain data from the Misteli lab (Cell 2019 – cited as ref.19 in the revised version). Unfortunately, we do not know what the Reviewer means by Ref. 43 as paper contains only 33 references in the previous version.

Page 4 in the revised main text contains a discussion of a few papers (including the one by Buckle et al) emphasizing the importance of heterogeneity. We do want to add that, to our knowledge, our work represents the first theoretical approach to analyze and predict the outcomes of the experiments.

2. A key interesting contribution of the paper is the fit of FISH data from Rao et al to get the fraction of populations with or without loops (Fig. 4). However I'd say it is typically difficult to discriminate between a FISH distribution which is uniform and one which has multiple components/subpopulations. In the current case the fit is only possible as far as I can see because there are only two assumed subpopulations. What would happen if there are additional loops within the loop anchors (nested loops, or combination of nested and consecutive loops)? there I would expect the combinatorial possibilities for each loop would need to be considered rendering fitting impossible/not too meaningful. In other words this method is mainly usable only for relatively small CTCF loops which do not have further loops nested in their interior, and I am not sure whether this is the rule or the exception in the data. Can the authors comment more on this and on how this discussion would change if more than two populations are relevant?

Response: This is a great question, which occurred to us as well during the course of this work. Two comments: (1) In order to worry about the distinct possibility of multiple subpopulation the data has to be more extensive and quite good as the Reviewer suggests. In the absence of such data we took the Occam's razor approach to analyze the Rao data set successfully. (2) However, in principle nothing prevents us from considering higher order heterogeneity and the associated architecture. In particular, there could be a distribution of populations, as the Reviewer correctly suggests. It turns out that the theory can be generalized. More importantly, but it proves to be necessary to consider a distribution of populations when dealing with the data in Ref. 19. We accomplished this task. The theory is described in the Supplementary Note 6. The procedure to solve the equations is given in the Supplementary Note 8. Examples of successful **quantitative** predictions of the findings by Misteli are given in Fig. 5 (for 6 loci

pairs) and the distributions for the 212 loci pairs are provided in Fig. 6 of the main text. Finally, in the Supplementary Fig.8 we provide complete analyses of the experimental data for all 212 loci pairs.

3. In the discussion the authors mention dynamic loop extrusion as a possibility to lead to a heterogeneous population but this would be multiple rather than two subpopulations, right? Also, they correctly point to [24,25] to say that cohesin/CTCF has finite lifetimes on DNA. While cohesin residence time is 20 min (approx) the work of Ref. [25] suggests CTCF gets off in a time of order 1 minute, over which it is difficult to believe that extrusion can processively lead to loops of 100 kbp or more. Therefore I am not sure that citing [25] is a convincing argument to say that dynamic loop extrusion can be used to explain the HiC FISH paradox/chromosome conformation heterogeneity. If the authors agree I suggest they rephrase this next-to-last paragraph before the Conclusions sections.

Response: Yes, the dynamic loop extrusion could lead to more than two subpopulations. We have added a sentence in the section. On page 11 in the revised main text, we add a sentence,

"At any given time, there would be many subpopulations each characterized by a distinct set of loops in the chromosome."

The issue of dynamics in loops, mediated by CTCF/cohesin, is very important as the Reviewer correctly points out. In pictorial terms the scenario for CTCF binding and unbinding would correspond to the schematic Figure 5b in the original version of main text (Figure 7b in the revised main text). In a theory that would include dynamics. Thus, the kinetics must be fully accounted for, no doubt. However, here we have concerned ourselves with equilibrium aspects of organization, and therefore the short lifetimes would further contribute to dynamic heterogeneity (Fig. 5b in the original version or Fig,7b in the revised version). We have rewritten the next-to-last paragraph before the conclusion section. On page 11, it now reads,

"Loop extrusion model [32–34] is another possible origin of dynamic heterogeneity. In the loop extrusion model, it is thought that cohesins extrude loops along the chromosome fiber, which could detach stochastically. At any given time, there would be many subpopulations, each characterized by a distinct set of loops in the chromosome. Indeed, our analyses of the most recent high throughput optical imaging data lends credence to the notion that multiple subpopulations in chromosomes arise because of massive dynamic heterogeneity.."

4. On page 7 a reference appears as ? to the chromosome copolymer model. Also, can the authors give a bit more details here?

Response: We apologize for the inappropriate ?. We fixed it. In this version, we also provide details of how to extract g and δ values. On page 8 in the revised main text, it now reads,

The value of g is inferred from the scaling between mean spatial distance $\langle R \rangle$ and contact probability P , $P \propto \langle R \rangle^{-(3+g)}$. The value of δ is computed as $\delta = 1/(1 - \nu)$ where ν is inferred from scaling $\langle R(s) \rangle \propto s^\nu$ where s is the genomic distance.

Reviewer #3 (Remarks to the Author):

Summary:

This paper designs a new Generalized Rouse Chromosome Model to explain the discrepancy between the chromosomal distance measured by FISH data and the chromosomal contact probability derived from the Hi-C data. It rigorously formulates the mathematical relationship between the distance and contact probability of loci and convincingly proves that the heterogeneity in the cell population is the factor causing the discrepancy. And it also presents an approach to calculate the fraction of sub cell populations from data. The theory agrees the experimental data well. The theory is successfully applied to the CTCF-mediated chromatin loops to calculate their distance. Overall, the theory is innovative and sound and the reasoning is logical. The theory is very useful for deriving the distance information from Hi-C and FISH data in order to build 3D structures of chromosomes and genomes.

Response: We are naturally pleased with the high praise of this reviewer. It is much appreciated.

Minor comments:

(1) The citation in this sentence “Chromosome Copolymer Model (CCM) for chromosomes [?]” is missing.

Response: We thank the reviewer for noticing the error. We have fixed it.

REVIEWERS' COMMENTS:

Reviewer #1 (Remarks to the Author):

My previous comments have been addressed.

I appreciate the authors' exploration of available experimental data and the expansion to their model in the added figures 5 and 6.

I have a minor new comment.

Figure 6a looks interesting but it is unclear if any interesting biology is hidden behind that bundle other than the message of heterogeneity.

Would it be possible to replot it as a heatmap (with each row corresponding to a pair of genomic locations)? It might be advised to order the columns by the genomic distance between the two loci.

Reviewer #2 (Remarks to the Author):

The authors have done a good job in my view of responding to the reviewers' comments. As far as my points are concerned, I am happy with the reply and I confirm that I support publication of this interesting work. As a small point, I apologise for the typo in my previous report when referring to Ref. 43. I meant the work by L. Giorgetti and collaborator (now Ref. 13 in revised version), and a useful reference could be additionally L. Giorgetti et al., Predictive polymer modeling reveals coupled fluctuations in chromosome conformation and transcription, *Cell* 157, 950-963 (2014). I will leave to the authors as optional whether to include this as well and briefly discuss it prior to publication.

Response to Reviewer #1:

My previous comments have been addressed.

I appreciate the authors' exploration of available experimental data and the expansion to their model in the added figures 5 and 6.

We are happy that this reviewer is satisfied with our analysis of experimental data.

I have a minor new comment.

Figure 6a looks interesting but it is unclear if any interesting biology is hidden behind that bundle other than the message of heterogeneity.

Would it be possible to replot it as a heatmap (with each row corresponding to a pair of genomic locations)? It might be advised to order the columns by the genomic distance between the two loci.

There are many ways of represented the results in Fig 6a. It is possible to plot Figure 6a as a heatmap but we find it does not give any more insights into the heterogeneous aspects than what Figure 6a already conveys. Because the data can be readily calculated using theoretical considerations, any reader can generate them and represent the data in any form. Hence, we decide that we will keep the figure 6a as it is. Indeed, understanding the biological significance of heterogeneity, which both experiments and theory are starting to point out, will be the focus in our future studies.

Response to Reviewer #2:

The authors have done a good job in my view of responding to the reviewers' comments. As far as my points are concerned, I am happy with the reply and I confirm that I support publication of this interesting work.

We are happy that this reviewer recommends publication of our paper.

As a small point, I apologise for the typo in my previous report when referring to Ref. 43. I meant the work by L. Giorgetti and collaborator (now Ref. 13 in revised version), and a useful reference could be additionally L. Giorgetti et al., Predictive polymer modeling reveals coupled fluctuations in chromosome conformation and transcription, Cell 157, 950-963 (2014). I will leave to the authors as optional whether to include this as well and briefly discuss it prior to publication.

We thank the referee for the suggestion. Because this excellent paper does not have a direct bearing in our present study for now we decided that it is not necessary to include it in the current.